# The Increase in the Elastic Range and Strengthening Control of Quasi Brittle Cement Composites by Low-Module Dispersed Reinforcement: An Assessment of Reinforcement Effects

**DOI:** 10.3390/ma14020341

**Published:** 2021-01-12

**Authors:** Dominik Logoń, Krzysztof Schabowicz, Maciej Roskosz, Krzysztof Fryczowski

**Affiliations:** 1Faculty of Civil Engineering, Wrocław University of Science and Technology, Wybrzeże Wyspiańskiego 27, 50-370 Wrocław, Poland; krzysztof.schabowicz@pwr.edu.pl; 2Faculty of Mechanical Engineering and Robotics, AGH University of Science and Technology, aleja Mickiewicza 30, 30-059 Kraków, Poland; mroskosz@agh.edu.pl; 3Faculty of Energy and Environmental Engineering, Silesian University of Technology, ul. Akademicka 2A, 44-100 Gliwice, Poland; krzysztof.fryczowski@polsl.pl

**Keywords:** quasi-brittle cement composites, low-module polypropylene fibres, elastic range

## Abstract

This paper presents the possibility of using low-module polypropylene dispersed reinforcement (E = 4.9 GPa) to influence the load-deflection correlation of cement composites. Problems have been indicated regarding the improvement of elastic range by using that type of fibre as compared with a composite without reinforcement. It was demonstrated that it was possible to increase the ability to carry stress in the Hooke’s law proportionality range in mortar and paste types of composites reinforced with low-module fibres, i.e., V_f_ = 3% (in contrast to concrete composites). The possibility of having good strengthening and deflection control in order to limit the catastrophic destruction process was confirmed. In this paper, we identify the problem of deformation assessment in composites with significant deformation capacity. Determining the effects of reinforcement based on a comparison with a composite without fibres is suggested as a reasonable approach as it enables the comparison of results obtained by various universities with different research conditions.

## 1. Introduction

The development of cement composites results in an increase in compressive strength without significantly improving bending strength [1,2]. The brittleness of cement composites causes rapid destruction, which is particularly disadvantageous in high-strength structures. Researchers have attempted to limit the brittleness of cement composites by trying to increase the flexural or bending strength using various fibre reinforcements [1,2,3,4,5,6,7,8,9,10,11,12,13,14,15,16,17,18,19,20,21,22,23,24,25,26,27,28,29,30,31,32,33,34,35,36,37,38,39]. Fibre reinforcement requires good rheological properties of the mix (for random dispersion), which determines a higher amount of cement. Short reinforcement controls the effect of multi-cracking, and longer reinforcement improves the toughness and strength [3,20,24]. The best results with respect to flexural strength and toughness have been achieved with a high-strength matrix and fibres [37].

High strength and high Young’s modulus reinforcement (steel fibre E = 210 GPa and carbon fibres E = 30–300 GPa) can be applied to increase stress corresponding to first crack appearance [29,30,31,32,33,34,35]. Improvement of the elastic range by means of low-module reinforcement is difficult to achieve, and therefore hybrid reinforcement is frequently used, with both high and low extension strength and Young’s modulus, and with different lengths [4,17,39]. High-module reinforcement controls the elastic range, while low-module reinforcement controls the deflection range after exceeding f_cr_ [5,6,7,8,9,10,11,12]. Existing publications do not indicate the effectiveness of applying low-module fibres for improving Hooke’s law proportionality range, which limits their use in construction materials for controlling the deformation and crack propagating process [13,15,16].

In this paper, we take into account the possibility of improving elastic range by using the application of low-module polypropylene reinforcement. Such fibres are commonly used, and a number of papers have been written with respect to their use. However, the existing papers do not focus on the possibility of using low-module polypropylene reinforcement to improve Hooke’s law proportionality range and obtain ESD composites (E, elastic range; S, strengthening control; and D, deflection control). These effects reduce the brittleness of the cement composites and help to avoid the catastrophic destruction process [13,14,15,16,17,18,19,20].

There are different methods of calculating flexural toughness, for example, the JSCE method, the ASTM C 1018 and the EN 14651 standard consistent with RILEM recommendations [40,41,42,43,44,45]. There is no one universal method to accurately describe and compare reinforcing ESD effects. The existing norms and standards describing these effects seem to be insufficient. This makes the assessment and comparison of the obtained effects difficult. Despite a number of formulas for the calculation of the reinforcement effect, new methods and modifications of the existing ones are still being proposed. Various papers have shown that rheological properties influence mechanical properties, especially with respect to concrete composites, whose rheological properties are much worse than those of mortar or paste composites [18,25].

According to a literature review, previous works have not shown the possibility of significantly increasing stress in the elastic range with low-module polypropylene fibres in structural elements as compared with matrix (unreinforced composite) [24,28,36,38].

In this paper, damaged composites were obtained with significant deflection and flexural strength that equalled or exceeded the strength corresponding to the first crack. Some of the effect had already been presented in our own works [21,22,23,24,25,26,27] for small beams of cement composites with synthetic structural polypropylene fibres [28] but not with respect to matrix (unreinforced composite).

In this paper, we focus on the limitation of the catastrophic destruction process by means of ESD effects in structural elements. The main goal of the paper was to improve the elastic range of cement composites by means of low-module fibres, which required the introduction of the maximum volume of dispersed reinforcement. That effect was obtained with the maximum volume of fibres for mortar, V_f_ = 3%, and for paste, V_f_ = 6%.

## 2. Materials and Methods

### 2.1. Materials Used for Tests

The following materials were used for the preparation of the cement composites: Portland cement (c) CEM I 42.5R (Górażdże cement plant, Górażdże, Poland) silica fume (10% c), fly ash (20% c), superplasticizer (SP, Sika company, Baar, Switzerland) tap water (w), and w/binder = 0.35. The sand used in the research is sold as sand for the production of ordinary concrete. The grain size distribution of the sand was 0–2 mm.

The composites were reinforced with randomly dispersed fibres (Figure 1 and Table 1) and synthetic structural polypropylene fibres (compliance with ASTM C 1116), specific weight 0.91 kg/dm^3^, flexural strength f_t_ = 620–758 MPa, E = 4.9 GPa, l = 54 mm, equivalent diameter 0.48 mm, and l/d = 113.

### 2.2. Preparation of Specimens for Tests

The specimens MV_f_3% and ZV_f_6% were reinforced with the maximum volume possible to disperse polypropylene fibres. All tested samples were demoulded and notched. Each beam was turned by 90° and cut to the depth of 30 mm (cut width 3 mm).

Components were mixed in the concrete mixer, and then used to mould samples. Beams (150 mm × 150 mm × 600 mm) were cast in slabs, and then cured in water at 20 ± 2 °C. After 180 days of ageing, the beams were prepared for the bending test, Figure 2. Figure 2a presents a sample prepared for the four-point bending test.

### 2.3. Description of the Test Stand

Four-point bending tests were carried out on the testing machine with closed-loop servo control displacement. The load-deflection curves (Figure 3, Figure 4, Figure 5, Figure 6, Figure 7, Figure 8 and Figure 9) were obtained according to ASTM C 1018, but the test was based on the measurement of the displacement of crosshead. The following data was obtained:-Tensile strength at bending f_max_ (MOR, the modulus of rupture), tensile strength at first crack f_cr_ (LOP, the limit of proportionality);-The characteristic points on the load-deflection curve, f_x_(F_x_-load, ε_x_-deflection, and W_x_-energy);-Energy (work) as proportional to the area under the load-deflection curve up to the characteristic point.

Additionally, deflection was recorded by means of two LVDT sensors located as in Figure 2. During the test, the bending load and deflection of the specimen were measured. The testing procedure corresponded to the requirements of the ASTM C 1018 standard.

The ESD reinforcement effect (i.e., elastic range, strengthening control, deflection control) is presented by characteristic points f_x_ and areas A_X_ under he load-deflection curve, Figure 3).

## 3. Test Results

The load-deflection curve for mortar MV_f_0% is presented in Figure 4. Additionally, the results presented earlier [24] for concrete without fibres and with the maximum volume content of the same fibres V_f_ = 2% are included (insignificant increase in the elastic range of concretes with the maximum volume of fibres was obtained). A typical load-deflection correlation for cement composites without reinforcement was obtained, with catastrophic destruction process (deflection as displacement of crosshead (mm)). The figure presents data corresponding to the maximum ability to carry stress f_cr_ = f_max_ = MOR and deformation capacity d_x_. The specimen is a reference for the other tested composites.

Figure 5 presents mortar with 2% of fibres. Characteristic points f_cr_, f_max_, f_d_, have been determined, which enable the identification of the following areas: elastic range A_E_ = 6.8 J, strengthening control A_S_ = 52.9 J, deflection control A_D_ = 83.1 J, and propagation area A_E_ is not significantly larger than A_E_ of the specimen without reinforcement. In addition, LOP, MOR, and d_x_ have been determined.

Mortar with the maximum possible volume of fibres is presented in Figure 6. The obtained results indicate a significant improvement of the properties of ESD composites in the following areas: elastic range A_E_ = 22.0 J, strengthening control A_S_ = 134.1 J, deflection control A_D_ = 130.2 J, and propagation area A_S_ with 3% of fibres is larger than A_D_. Additionally, LOP and MOR were determined. We found that reinforcement significantly contributed to the increased deformation capacity in the elastic range d_x_ = 15.8.

Figure 7 shows a paste specimen ZV_f_6% with the maximum possible volume of fibres V_f_ = 6%. The best results were obtained regarding the ability to carry stress in the elastic range A_E_ = 184.4 J, strengthening control area A_S_ = 134.1 J, deflection control area AD = 271.3 J, and propagation area A_P_. Significant improvement of the ability to carry stress has been achieved for LOP and MOR and a slight improvement of deformation capacity d_x_.

The compilation of all the tested specimens (load-deflection curves) is presented in Figure 8. The curves illustrate the scale of obtained ESD effects as compared with the matrix (mortar without fibres). For structural reasons, it is important to improve stress in Hooke’s law area and not those corresponding to f_max_. As the presented curves show, the specimens with the content exceeding V_f_ = 2% may show significant ESD effects.

Figure 9 shows specimen ZV_f_6% after the four-point bending tension test. The figure presents two sensors measuring deflection relative to the neutral axis. It should be noted that there are significant differences regarding the displacement of the cut edges in the case of considerable deformations of the specimen. The visible differences result in significant differences in the measurements of ESD composites’ deflection.

Table 2 presents a compilation of the results of the four-point bending tension test of the tested specimens (load, deflection, absorbed energy, LOP, MOR, and d_x_) with the data that correspond to various characteristic points f_x_ (f_cr_, f_max_, and f_d_).

In order to compare the results with results obtained in other research centres, the results are compared with a reference matrix (specimen of mortar without reinforcement MV_f_0%). The obtained results indicate multiple changes (improvement/deterioration of properties) as compared with the original composite, Table 3.

During the analysis of the results, significant discrepancies were found regarding the determination of deflection by means of crosshead displacement and by means of reference to the neutral axis of the specimen with respect to Hooke’s law proportionality range, point f_cr_. In the case of a beam made of paste with the maximum content of fibres ZV_f_6%, the following deflection was recorded relative to the neutral axis in point f_cr_, sensor 1, 0.774 mm and sensor 2, 0.879 mm, Figure 9 The average deflection was 0.827 mm, while the corresponding crosshead displacement was significantly larger and equalled 2.078 mm.

Table 3 proposes a method of comparing the results. The reference matrix was mortar without reinforcement (any other composite can be taken as a reference point).

## 4. Discussion

Increasing the ability to carry stress in the elastic range in ESD cement composites reinforced with low-module fibres is very limited (as compared with high-module reinforcement) and even impossible if the volume of fibres is low. The conducted tests show that with the content of fibres exceeding V_f_ = 2% in mortar cement composites there is such a possibility. Previous tests carried out on concrete composites indicated there was no such possibility with V_f_ = 2% [24], Figure 4. In the presented tests (Figure 8), we found that it was possible, if a large volume of dispersed reinforcement was introduced at the level of V_f_ = 3%. It was impossible in concrete composites [13,28], due to the deteriorating (along with the increasing fibre content) rheological properties of the mixes [25].

The best ESD effects were demonstrated in paste mixes (with the best rheological parameters as compared with mortars and concretes), which enabled the addition of the largest volume of fibres V_f_ = 6%, Figure 8. It should be noted that such a large content of dispersed reinforcement makes it difficult to form the mix, which makes it predestined for use in the prefabrication of cement composites. The use of smaller quantities of the reinforcement (which improved the rheological parameters of the mixes) resulted in decreased ESD effects in those composites. The obtained results indicate the possibility of using this type of reinforcement, for example, in prefabricated thin-walled composites such as building facade cladding panels. ESD effects limit the catastrophic destruction process of these composites (e.g., earthquake or mechanical damage). Paste with the maximum fibres volume was proposed as the ESD composite instead of mortar, because it was possible to introduce twice as many fibres into the paste composite, which resulted in a significant improvement of the elastic range of those composites, which was the main goal of this paper.

In the presented test results, emphasis was placed again on the need to compare the obtained effects f_cr_, f_max_, f_d_ with the parameters of a reference matrix (without reinforcement). The ESD composites should show greater ability to carry stress in the elastic range, strengthening control area, and a considerable range of deflections in the deflection control area. Assigning appropriate symbols to the tested composites enables the identification of their behaviour under load, and it is possible to describe them in detail on the basis of the characteristic points f_x_ (F_x_, load; ε_x_, deflection; W_x_, energy and areas A_X_, Figure 3).

The determination of the absorbed energy by means of ASTM 1018 (I_5_,I_10_,I_15_) and crack mouth opening displacement (CMOD) did not correlate with f_cr_ f_max_, and f_d_, rendering a description of the behaviour under load in various deflection areas impossible. We found that the values of deflection measured by means of crosshead displacement were considerably higher than those measured in relation to the neutral axis. The measurement of deflection in the elastic range in accordance with [40] is a good method of deflection assessment. We confirmed that the deflection of ESD composites outside the elastic range relative to the neutral axis could not be controlled, Figure 9. Despite the significantly larger deflections measured by means of crosshead displacement as compared with the measurement in relation to the neutral axis, the results should be considered to be a good basis for the assessment of the elastic range, assuming that those diagrams are made based on the relationship between linear force and deflection (the initial deflections and settlement on supports resulting in disproportionately larger deflections are not taken into consideration).

An attempt to standardise the symbols and description of the method of testing various building materials under load seems to be justified. The existing recommendations and guidelines focus on tests and symbols ascribed differently to different materials when describing the behaviour of materials under load, Figure 3.

The testing of materials under load also needs to be standardised in terms of the method of assessing the obtained results. The impact of a number of variables (size of the specimen, cutting, load application speed, humidity, etc.) often makes the results from tests obtained in various universities difficult to compare. Again, we suggest the possibility of assessing the results based on a comparison with composite without reinforcement, and therefore the results could be approximately compared with the results obtained at various universities, limiting the influence of the scale of specimens and testing methods on the obtained results, since the results would be assessed based on multiple improvements/deteriorations as compared with a reference matrix. The possibilities for comparing results are presented in Table 3 in relation to the four-point bending test, which requires further discussion.

## 5. Conclusions

We have demonstrated that it is possible to improve the ability to carry stress in Hooke’s law proportionality range in cement composites reinforced with low-module fibres if a large quantity of dispersed reinforcement exceeds V_f_ = 3%. That condition cannot be fulfilled in traditional concrete structures due to worse rheological parameters of the mix as compared with mortar or paste composites.

The best ESD effects were demonstrated in the elastic range (and additionally in strengthening and deflection control areas) in paste with the maximum volume of fibres V_f_ = 6%.

We suggest that there is a need to assess the obtained effects f_cr_, f_max_, f_d_ based on a comparison with the parameters of matrix (specimens without reinforcement), in order to identify quasi-brittle composites as ESD, with increased ability to carry stress in the elastic range, strengthening control area, and a considerable range of deflections in the deflection control area.

The values of deflection measured by means of crosshead displacement were demonstrated to be considerably higher than those measured in relation to the neutral axis. However, it was difficult to assess the ESD effects in relation to the neutral axis and crack mouth opening displacement.

## Figures and Tables

**Figure 1 materials-14-00341-f001:**
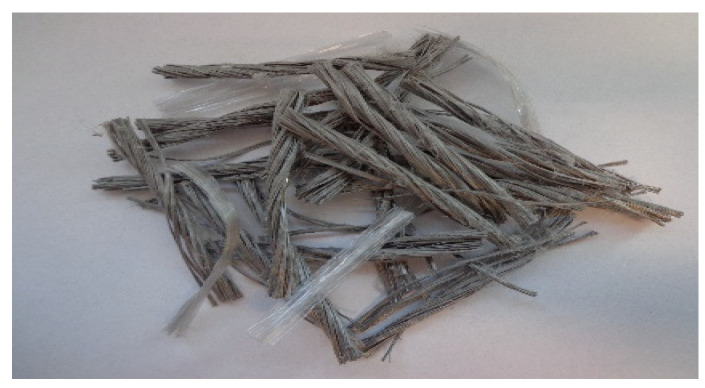
Synthetic structural polypropylene fibres l = 54 mm.

**Figure 2 materials-14-00341-f002:**
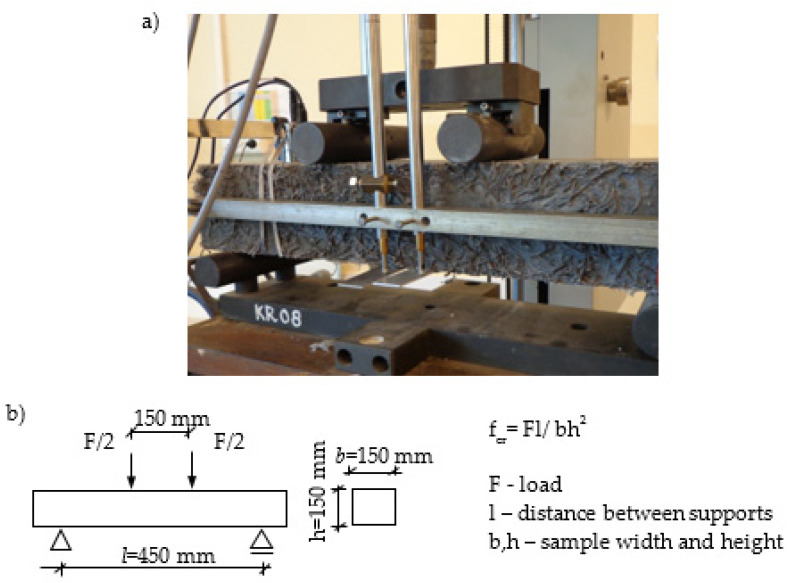
Four-point bending test. (**a**) Specimen before test, paste V_f_ = 6%; (**b**) Diagram of the test.

**Figure 3 materials-14-00341-f003:**
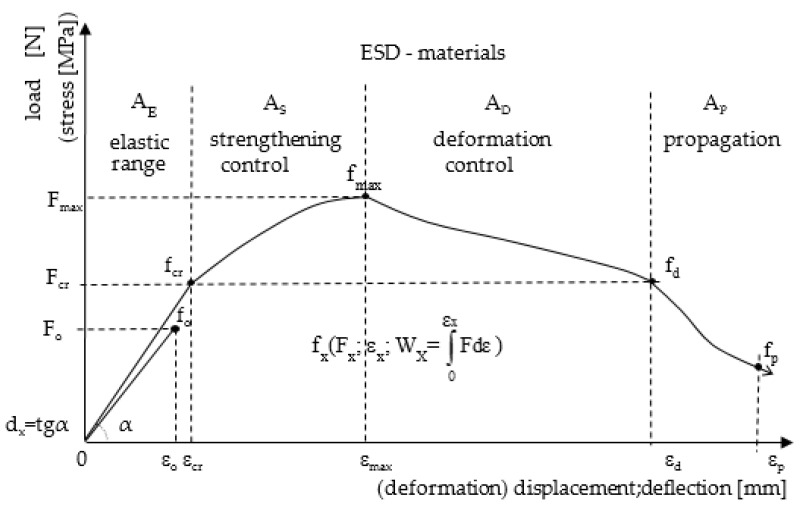
ESD composites depending on load deflection. Areas: A_E_, elastic range; A_S_, strengthening control; A_D_, deflection control; A_P_, propagation.

**Figure 4 materials-14-00341-f004:**
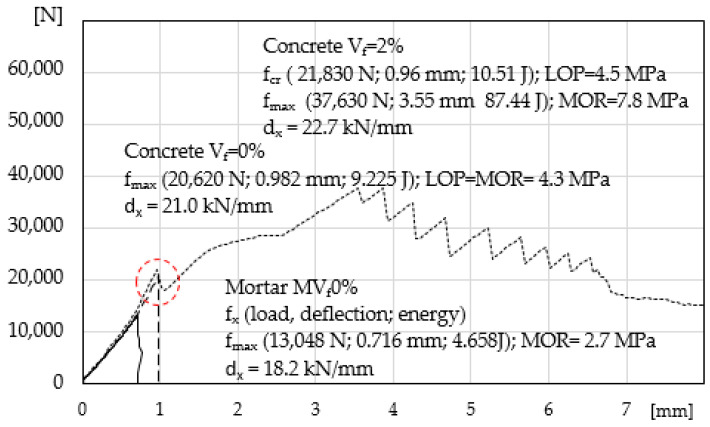
Load-deflection curve in the four-point bending test, mortar MV_f_0% (matrix) and concrete without fibres V_f_ = 0% and with V_f_ = 2% fibres.

**Figure 5 materials-14-00341-f005:**
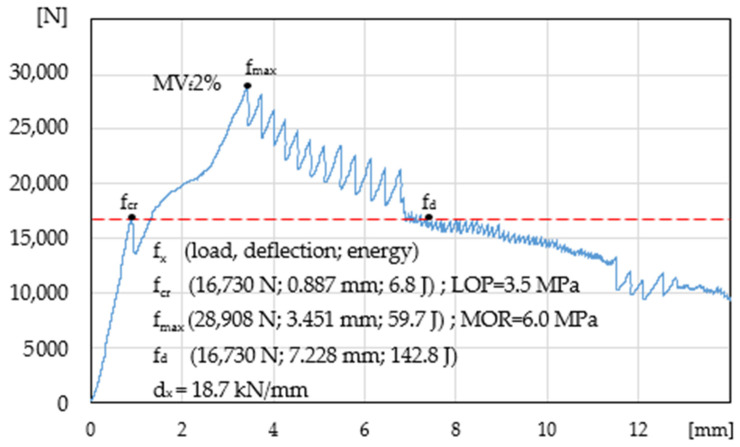
Load-deflection curve in the four-point bending test, mortar MV_f_2%.

**Figure 6 materials-14-00341-f006:**
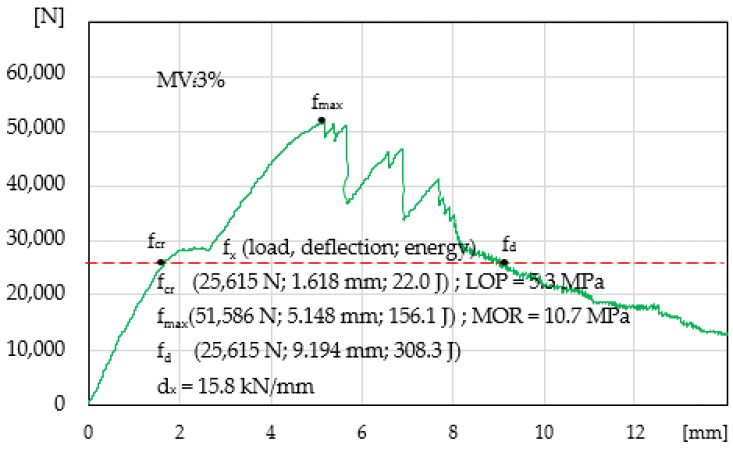
Load-deflection curve in the four-point bending test, mortar MV_f_3%.

**Figure 7 materials-14-00341-f007:**
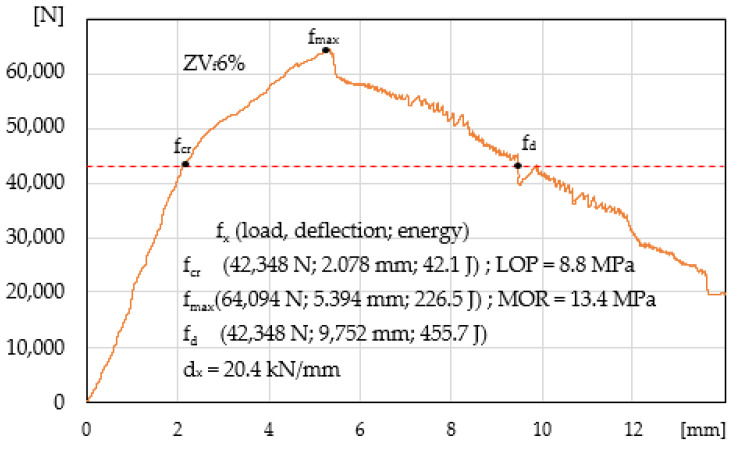
Load-deflection curve in the four-point bending test, paste ZV_f_6%.

**Figure 8 materials-14-00341-f008:**
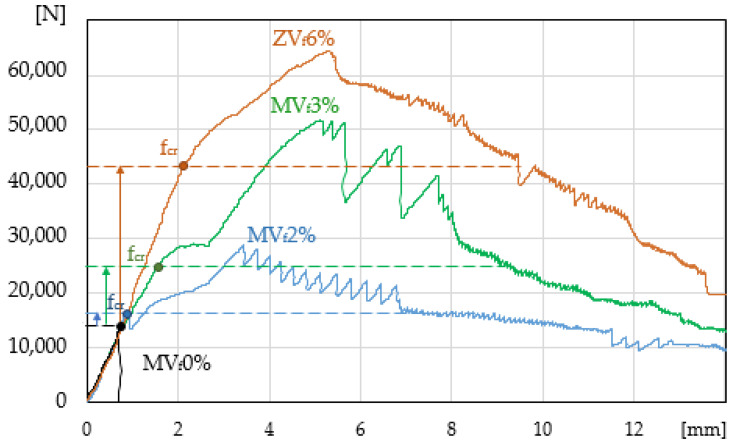
Load-deflection curve in the four-point bending test, comparison of samples.

**Figure 9 materials-14-00341-f009:**
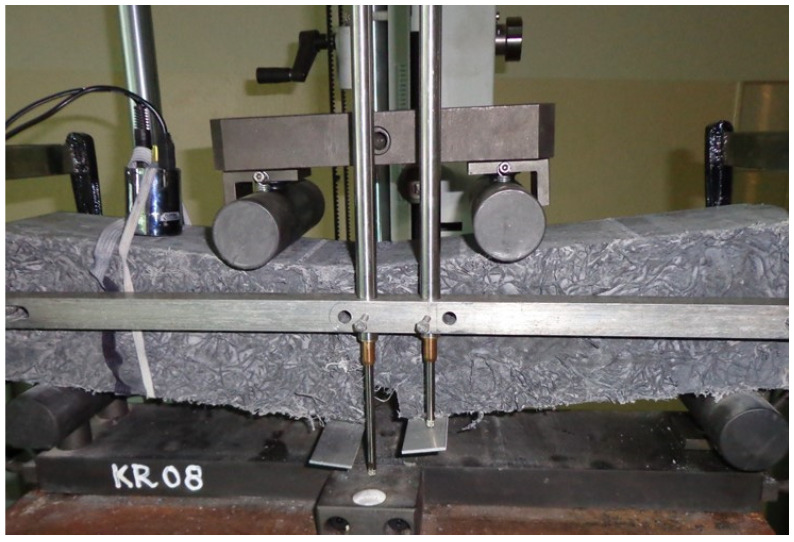
The four-point bending test, paste ZV_f_6% after the test.

**Table 1 materials-14-00341-t001:** Tested specimens.

Symbol	Specimen	Cement:Sand (Volume)	V_f_ [%]
MV_f_0%	mortar	1:4.5	0
MV_f_2%	mortar	1:4.5	2
MV_f_3%	mortar	1:4.5	3
ZV_f_6%	paste	-	6

**Table 2 materials-14-00341-t002:** Tested specimens, a compilation of data in relation to characteristic points f_x_.

Composite	Load	Deflection	Energy	LOP	MOR	d_x_
	(N)	(mm)	(J)	(MPa)	(MPa)	(kN/mm)
MV_f_0%						
f_cr_ = f_max_	13,048	0.716	4.7	2.7	2.7	18.2
MV_f_2%						
f_cr_	16,730	0.887	6.8	3.5		18.7
f_max_	28,908	3.451	59.7		6.0	
f_d_	16,730	7.228	142.8			
MV_f_3%						
f_cr_	25,615	1.618	22.0	5.3		15.8
f_max_	51,586	5.148	156.1		10.7	
f_d_	25,615	9.194	308.3			
ZV_f_6%						
f_cr_	42,348	2.078	42.1	8.8		20.4
f_max_	64,094	5.394	116.5		13.4	
f_d_	42,348	9.752	455.7			

**Table 3 materials-14-00341-t003:** The tested specimens, a compilation of the obtained results as compared with a reference matrix.

Composite	Load (N)	Deflection (mm)	Energy (J)	LOP (MPa)	MOR (MPa)	d_x_ (kN/mm)
matrix MV_f_0%						
f_0_ = f_cr_ = f_max_	13,048	0.716	4.7	2.7	2.7	18.2
MV_f_2%/MV_f_0%						
f_cr/0_	1.28	1.24	1.45	1.30		1.03
f_max/0_	2.22	4.82	12.70		2.22	
f_d/0_	1.28	10.09	30.38			
MV_f_3%/MV_f_0%						
f_cr/0_	1.96	2.56	4.68	1.96		0.87
f_max/0_	3.95	7.19	33.21		3.96	
f_d/0_	1.96	12.84	65.60			
ZV_f_6%/MV_f_0%						
f_cr/0_	3.25	2.90	8.96	3.26		1.12
f_max/0_	4.91	7.53	24.79		4.96	
f_d/0_	3.95	13.62	96.96			

## Data Availability

Not applicable.

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
