# Peer review of "The Increase in the Elastic Range and Strengthening Control of Quasi Brittle Cement Composites by Low-Module Dispersed Reinforcement: An Assessment of Reinforcement Effects"

_materials, 2021, doi:10.3390/ma14020341_

Round 1

Reviewer 1 Report

See the file attached

Author Response

We are deeply grateful to the Reviewer for the effort put in the review of our paper.

(attachment)

Reviewer 2 Report

The research seems good but it needs improvement interms of a scientific paper. Following are comments:

  1. Avoid short paragraphs e.g. lines 56-58, one sentence paragraph.
  2. Labelling should be done in figure 2.
  3. Figure 3 may be merged with figure 2 as b.
  4. Figures 5 to 9 should be in one figure as a to e.
  5. Rationale behind variable selection should be mentioned at an appropriate location.
  6. A paragraph should be added in discussion section about practical implementation of current research in real life projects.

Author Response

(The authors gave the same response as above.)

Reviewer 3 Report

This research might be published if some aspects are improved. In my opinion, this paper is well structured and results are clearly presented, however some sections could provide more detail and discussion should be improved. It seems that the use of higher amount of fibres will present a better performance in terms of load-deflection curve, so results obtained were expected. Please highlight the novelty of this research and its contribution to the knowledge in this field. 

In order to improve the quality of this study, take into account the following comments and suggestions: 

- Authors could provide properties of sand used in this research. Only fibres are described in detail.

- Did you perform any test to obtain basic concrete properties such as mechanical strengths, density, …? This could be interesting for the research and it would improve the quality of discussion of results.

- Three mortars present the same composition and the only variable is the amount of fibres used, however the sample made of paste present a different amount of fibres. Why do you propose the use of a sample made of paste instead of mortar? How do you compare this results to those obtained from samples of mortar if there is any common parameters? Please justify this issue in the manuscript.

- Authors could include the comparison of experimental results to predictions of different codes.

Author Response

(The authors gave the same response as above.)

Round 2

Reviewer 3 Report

Please include your answers in the manucript. You only justify tour response, however no modifications were made in the manuscript according to my suggestions. 

Regarding the question about the use of paste specimens, please clarify your answer. 

Author Response

We are deeply grateful to the Reviewer for the effort put in the review of our paper.
